# Resource-Adaptive Federated Learning with All-In-One Neural Composition

**Yiqun Mei   Pengfei Guo   Mo Zhou   Vishal M. Patel**
Johns Hopkins University
{ymei7,pguo4,mzhou32,vpatel36}@jhu.edu

## Abstract

Conventional Federated Learning (FL) systems inherently assume a *uniform* processing capacity among clients for deployed models. However, diverse client hardware often leads to varying computation resources in practice. Such *system heterogeneity* results in an inevitable trade-off between model complexity and data accessibility as a bottleneck. To avoid such a dilemma and achieve *resource-adaptive federated learning*, we introduce a simple yet effective mechanism, termed *All-In-One Neural Composition*, to systematically support training complexity-adjustable models with flexible resource adaption. It is able to efficiently construct models at *various* complexities using *one* unified *neural basis* shared among clients, instead of pruning the global model into local ones. The proposed mechanism endows the system with unhindered access to the full range of knowledge scattered across clients and generalizes existing pruning-based solutions by allowing soft and learnable extraction of low footprint models. Extensive experiment results on popular FL benchmarks demonstrate the effectiveness of our approach. The resulting FL system empowered by our All-In-One Neural Composition, called FLANC, manifests consistent performance gains across diverse system/data heterogeneous setups while keeping high efficiency in computation and communication.

## 1 Introduction

The success of deep learning is greatly empowered by the large datasets [1, 2]. However, when data is scattered across a vast number of edge devices, and subject to privacy regulations, deep learning will be unrealistic. To this end, Federated Learning (FL) [3, 4] provides a decentralized solution, enabling a number of participants to jointly train a model without exchanging any raw data. Due to such privacy-preserving property, FL is favorable and widely adopted in many applications [5, 6, 7, 8, 9, 10], including but not limited to face recognition [11], autonomous driving [12] and next-word prediction [13].

When designing federated learning systems [14, 15, 16, 3, 17, 18], it is commonly assumed that the local models share the same architecture as the global one. However, real-world deployments can rarely satisfy this *ideal* assumption. Instead, participants *in the wild* often equip with diverse devices (*e.g.*, Internet of Things (IoT) devices, mobile phones, tablets and personal computers), which greatly vary in computation resource budgets (including computation capacity, memory, storage, and network bandwidth). Further, even if participants equip with identical devices, the resource budgets may still vary due to *e.g.*, other irrelevant programs consuming an arbitrary portion of the resource, or reduced CPU frequency due to power-saving mode. This ubiquitous presence of **system heterogeneity** becomes a major bottleneck for real-world FL applications, as it leads to an inevitable trade-off between model complexity and data accessibility – service providers have to either sacrifice model capacity to enable training on indigent devices, or exclude them along with their unique data. Predictably, in such a dilemma, both choices will eventually harm the performance of the resulting model.

36th Conference on Neural Information Processing Systems (NeurIPS 2022).

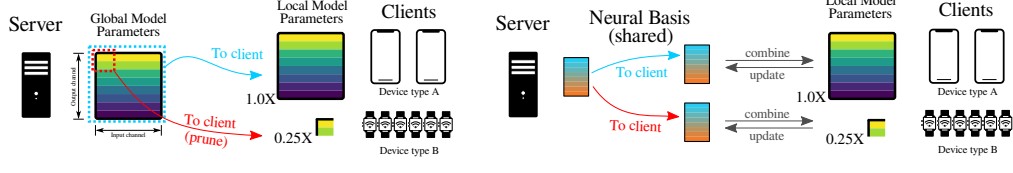

|  (a) HeteroFL and FjORD | (b) FL with All-In-One Neural Composition |

Figure 1: Comparison of different system-heterogeneous strategies. Schematics are demonstrated using one hidden layer on two different types of devices A and B. (a) HeteroFL [19] and FjORD [20] prune excessive parameters to adapt to a smaller device. In this case, a large portion (more than 90%) of parameters between $0.25\times$ and $1.0\times$ nets cannot learn from the data on devices type B. (b) our approach allows all parameters to leverage and learn from the full range of data.

The existing FL methods with heterogeneous clients, *e.g.*,HeteroFL [19] and FjORD [20] propose to modify the width of deep neural networks, *i.e.*, to extract low-footprint sub-models by pruning excessive channels. However, we find their performance improvement is inconsistent, which could be even worse than naively deploying the weakest network with the vanilla FedAvg [15]. As shown in Figure 1, we use an FL setting with two types of devices A and B, where the high-end device type A takes up 25% clients and can run the full model ($1.0\times$ in terms of the model width), while the low-end device type B takes up the majority (75%) of participants but can only afford $0.25\times$ of the full model. A counter-intuitive result is observed with standard ResNet-18 [21] and CIFAR10 [22] dataset: the Top-1 accuracy of both methods **decrease** from 86% to 84% when **increasing** model capacity from $0.25\times$ to $1.0\times$. Based on this, we conjecture that the extra $0.75\times$ network capacity does not benefit from all participants' knowledge. Even if all participants contribute to the same global model, pruning in fact limits the contribution of weakest devices to merely the $0.25\times$ network, leaving the majority of parameters under-optimized. This can be even more severe when there is *data heterogeneity*, where inadequately trained parameters may lead to poor generalization against unseen classes, and hence worse the overall performance.

*Given heterogeneous devices, how to unleash the full potential of resource capacity without sacrificing data accessibility?* In this paper, we propose All-In-One Neural Composition, a new scheme for run-time resource-adaptive model construction, which formulates networks at different capacities (thus footprint and data transfer cost) as linear combinations of *one* unified set of parameters, namely *neural basis*. It is designed to be compact, and regularized to be linearly independent, and hence is efficient in training and communication, as well as more faithful model constructions. With such scheme incorporated, the resulting FL framework (called FLANC) provides systematic support for training complexity-adjustable neural networks, and thus enables efficient and more effective *resource-adaptive federated learning*. Different from pruning-based methods [19, 20], it makes knowledge contribution from clients to model parameters unhindered from both computational resource constraints and model architectural complexity. Our method can be seen as a generalization of existing solutions by extending hard-coded weight masks to soft and learnable extractions of lower footprint models. In addition, similar to previous approaches [19, 20], original models can be pre-composed for inference and executed as normal networks without any run-time overhead.

To validate our approach, we conduct extensive experiments under both statistical data heterogeneity (IID and non-IID distribution) and system heterogeneity (static and dynamic) settings. We evaluate our method on commonly used datasets, *i.e.*, Fashion-MNIST [23], CIFAR10 and CIFAR100 [22] for image classification, as well as Shakespeare [24] for next-character prediction. Although the proposed mechanism is simple in its formulation, it is stable in training, while consistently and significantly outperforming the previous system-heterogeneous strategies [19, 20] across different architectures.

## 2 The Proposed Method

### 2.1 System Heterogeneity

As aforementioned, conventional FL systems assume a uniform processing capacity for all participants. Nevertheless, edge devices in the wild often equip with diverse hardware and may have very different computational resources and data transfer budgets. Thus, it is crucial to properly model system heterogeneity. When reflecting this concept to the deep networks, resource constraints can be

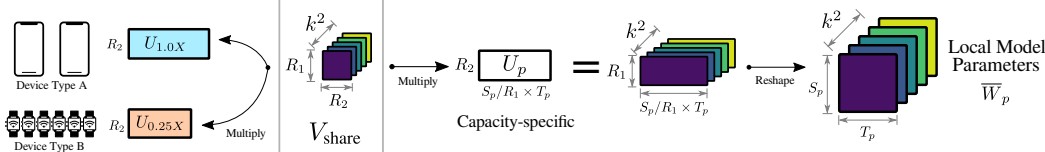

Figure 2: All-In-One Neural Composition using the shared neural basis $V_{\text{share}}$ and capacity-specific tensor $U_p$. Note, in the rightmost reshaping operation, $S_p = R_1 \times S_p/R_1$.

expressed in multiple aspects, such as depth (number of layers) and width (number of hidden channels). In this paper, we follow [19, 20] and choose to adjust width for resource adaptation. Compared to reducing depth, a reduction in width can more effectively reduce parameters and memory footprint during inference, which is beneficial to the edge devices. Meanwhile, the resulting networks belong to the same model class and share similar intrinsic characteristics, which is preferred for stabilizing training and model aggregation [19, 20, 25, 26].

We denote the ratio $p$ as a rescaling factor of the number of active channels in a layer $W \in R^{S \times T}$. Both the model size and computational cost of a slimmer $p$-width model will be reduced by $p^2$. To model device in the wild, one can assign each device $k$ with a maximum affordable capacity ratio $p_k$ according to its hardware configuration. Since it is impractical to exhaustively list all possible values of $p$ due to the diversity of edge devices[1], we consider a set of representative capacities $\mathcal{P}$ (e.g. $\{0.25, 0.5, 0.75, 1\}$) by pre-clustering participating devices into $|\mathcal{P}|$ resource groups according to their capacity.

## 2.2 All-In-One Neural Composition

Previous research suggests that the weights of fully-connected and convolution layers are usually over-parameterized, and lie on a low-rank subspace [27]. This makes it possible to express these layers in the low-rank tensor format using decomposition techniques [28, 29]. Given a convolution weight $W^{k^2 \times S \times T}$ with filter size $k$, input channel number $S$ and output channel number $T$, we can approximate it as $W = V \cdot U$ with $V \in R^{k^2 \times S \times R}$, $U \in R^{R \times T}$, and $R \leq T$. The factorization for fully-connected layer is equivalent as the $k = 1$ case. Such low-rank approximation is effective for model compression in order to reduce model size and computation cost [30, 31, 32, 33]. Similarly, we speculate federated learning with system heterogeneity could benefit from tensor decomposition.

Thus, we introduce All-In-One Neural Composition, that employs low-rank approximation to represent networks in different widths with a unified expression. For an arbitrary $p$-width network, we decouple its weight $W_p \in R^{k^2 \times Sp \times Tp}$ as a shared tensor $V_{\text{share}}$ and capacity-specific tensor $U_p$, *i.e.*,

$$W_p \approx V_{\text{share}} \cdot U_p \tag{1}$$

However, Eq. (1) cannot be directly used as varying $p$ not only changes the output channel but also rescales the input channel $S$ by $p$, which makes sharing $V_{\text{share}}$ of default shape $k^2 \times S \times R$ inapplicable. We address this issue by adjusting $V_{\text{share}}$ to be more fine-grained. We demonstrate this process in Figure 2. Specifically, given $V_{\text{share}}$ with shape $k^2 \times R_1 \times R_2$, we select $R_1$ from the common divisors of all possible input channel $Sp$ with $p \in \mathcal{P}$. Correspondingly, the capacity-specific tensor $U_p$ has the shape $R_2 \times (Sp/R_1 \times Tp)$. Their multiplication result is a reshaped weight matrix $\overline{W}_p \in R^{k^2 \times (R_1 \times Sp/R_1) \times T_p}$ and can be used for the original $p$-width network.

Eq. (1) in fact means to compose $W_p$ with a shared basis $V_{\text{share}}$ and coefficients $U_p$ through linear combination. Specifically, every weight fragment $W_{i,p} \in R^{k^2 \times R_1}$ of $\overline{W}_p$ is explicitly written as the weighted sum of $R_2$ bases $V_{\text{share}} = \{V_j | j \in \{1, \ldots, R_2\}\}$ with coefficients $U_{i,p}$, *i.e.*,

$$W_{i,p} = \sum_{j=1}^{R_2} u_{j,i,p} V_j \tag{2}$$

where $U_{i,p}$ is the $i$-th column of $U_p$, and $V_j \in R^{k^2 \times R_1}$ is the $j$-th basis vector in $V_{\text{share}}$.

---

[1]For example, there are 24,000 unique Android devices by 2015.

Eq. (2) reveals an outstanding property that the knowledge learned by $V_{\text{share}}$ can be propagated to all parameters. In the context of FL, this enables every parameter to access the full range of knowledge by training the basis $V_{\text{share}}$ on all devices. In addition, our formulation also reduces both computation and communication cost as the low-rank tensor format is inherently more efficient.

**Comparison to low-rank model compression.** Low-rank factorization is generally applied as a post-processing step to the pre-trained weights for model compression [34, 28, 30, 29]. Most methods produce a *single* compressed model and require access to the *labelled data* for fine-tuning. Hence, in the context of FL, the standard compression techniques neither address system heterogeneity nor reduce the footprint and data transfer cost, because they are only applied after the training process. Moreover, the requirement of fine-tuning is often infeasible for privacy-preserving applications. In contrast, our method is applied during the training process, and is able to flexibly construct *multiple* networks and can be stably learned from scratch without post-calibration.

**Comparison to pruning-based FL strategies.** Current system-heterogeneous strategies [19, 20] are pruning-based and can be viewed as a specific instantiation of Eq. (1), by setting $V_{\text{share}}$ to the largest weight $W_{p_{max}}$ and the coefficients $U_p$ to be a hard-coded mask with 0 and 1 entries, indicating whether a parameter is pruned or not. From this viewpoint, our method generalizes existing solution by allowing (1) a more compact basis $V_{\text{share}}$, which ensures high efficiency and reduces data transfer cost; and (2) learnable soft coefficients $U_p$, which makes both access to the full range of knowledge and adaptive utilization possible. Namely, unlike pruning-based methods [19, 20] where the excluded parameters cannot benefit from data on smaller devices, our approach enables *every* parameter at *all* width $p \in \mathcal{P}$ to leverage the *full* range of knowledge from *all* clients, no matter whether it can be trained on that device or not.

## 2.3 Orthogonal Regularization for Enhancing Representation Capacity

The low-rank tensor format of Eq. (2) can also be interpreted as adaptively searching for parameters from a low-rank subspace $\mathcal{S} = \text{span}\{V_{\text{share}}\}$. Spontaneously, if we enforce these basis vectors to be linearly independent, they can form a more expressive subspace and are capable to cover a broader range of knowledge. Nevertheless, according to our observation, this preferable property cannot be obtained implicitly with standard training objectives alone.

To this end, we introduce an extra orthogonal regularizer, which enforces the columns of $V_{\text{share}}^T$ to be mutually orthogonal with unit norm. We achieve this by reducing the $L_2$ norm of the difference between Gram matrix[2] and identity matrix. To train a $p$-width network, one can jointly optimize the orthogonal regularizer and the original classification objective, which can be expressed as:

$$\mathcal{L}_p = \mathcal{L}_{cls}(x, y; V_{\text{share}}, U_p) + \lambda \sum_{l=1}^{L} \|V_{\text{share}}[l] \cdot V_{\text{share}}[l]^T - I\|^2 \tag{3}$$

Here for simplicity, we use $V_{\text{share}}$ and $U_p$ to denote the collections of basis and coefficients over all layers. $\mathcal{L}_{cls}$ is the classification loss. $l$ is the layer index and $\lambda$ is a balancing parameter.

## 2.4 Federated Learning with All-In-One Neural Composition (FLANC)

In this subsection, we employ our composition technique into the FL framework to enable resource-adaptive federated training. We abbreviate our framework as "FLANC" for convenience.

As aforementioned, the first step of our framework is to obtain a feasible set $\mathcal{P}$ of device capacity by clustering participants into $|\mathcal{P}|$ groups based on their available resource. Meanwhile, the server also assigns each client $k$ with a corresponding capacity $p_k$ based on the clustering results. Next, the server initializes the shared basis $V_{\text{share}}^0$ as well as all capacity-specific coefficients $U_p^0$ with $p \in \mathcal{P}$. It is worth noting that our server does not need to explicitly maintain the global networks during training. The data transfer cost is further reduced with the efficient tensor format.

A communication round $t$ starts by determining a set of joining devices $\mathcal{K}_t$. After that, the server broadcasts the basis $V_{\text{share}}^t$ to each client $k \in \mathcal{K}_t$ along with the associated resource-specific coefficients $U_{p_k}^t$ for ad-hoc adaption. On the client side, each client updates the model with their private data for $N$ local iterations. For each training step, $p_k$-width network is adaptively constructed using

---

[2]To compute the Gram matrix, neural bases are first transposed into columns, *i.e.*, $V_{\text{share}}^T$

---

**Algorithm 1:** Federated Learning with All-In-One Neural Composition (**FLANC**)

---

**Input:** $V_{\text{share}}^0$ neural basis; $U_{p_1}^0 U_{p_2}^0, ..., U_{p_n}^0$ with $p_i \in \mathcal{P}$, $|\mathcal{P}|$ capacity-specific coefficients; $T$ communication rounds; $N$ local iterations

1  **for** $t \leftarrow 0$ **to** $T - 1$ **do**  // *Global communication rounds*
2      Server determines a set of joining devices $\mathcal{K}_t$;
3      Server broadcasts the neural basis $V_{\text{share}}^t$ and coefficients $U_{p_k}^t$ to each client $k \in \mathcal{K}_t$
4      **for** $k \in \mathcal{K}_t$ **do**  // *do in parallel among clients*
5          **for** $n \leftarrow 0$ **to** $N - 1$ **do**  // *local iterations for every client*
6              Client composes the $p_k$-width network using Eq. (1);
7              Client computes the loss objective using Eq. (3), and then updates $V_{\text{share}}^{t,k}$ and $U_{p_k}^{t,k}$;
8          **end**
9          Client uploads the updated neural basis and capacity-specific coefficients to the server;
10     **end**
11     Server collects the updates of the shared basis and coefficients using Eq. (4);
12 **end**

---

Eq. (1). Therefore, the basis $V_{\text{share}}$ can be optimized by all clients with their local data. At the end of training, the clients send the basis and coefficients back to the server.

The server finishes a communication round by updating the shared basis and coefficients using local training results from clients. In our approach, the basis is fully trained on all participating devices. On the other hand, capacity-specific coefficients are updated by the devices with the corresponding resource group. To account for this difference, we provide the following aggregation rule:

$$V_{\text{share}}^{t+1} = \frac{1}{|D_{\mathcal{K}_t}|} \sum_{k \in \mathcal{K}_t} |D_k| V_{\text{share}}^{t,k} \quad \text{and} \quad U_p^{t+1} = \frac{1}{|D_{\mathcal{K}_{t,p}}|} \sum_{k \in \mathcal{K}_{t,p}} |D_k| U_p^{t,k} \tag{4}$$

where $D_{K_t} = \cup_{k \in \mathcal{K}_t} D_k$ is the union of datasets of current participants. And $\mathcal{K}_{t,p} = \{k | k \in K_t, p_k = p\}$ is a subset of $\mathcal{K}_t$ with the device capacity equals $p$. The complete workflow of our method is summarized in Algorithm 1.

## 3 Experiments

**Datasets.** In this paper, we evaluate our method for image classification tasks on three popular datasets with increasing complexity: Fashion-MNIST [23], CIFAR10 and CIFAR100 [22]. Fashion-MNIST is a relatively simple dataset containing 60,000 examples of 10 classes. CIFAR10 and CIFAR100 are the common classification benchmarks with 50,000 training images with 10 and 100 classes respectively. CIFAR100 is the most challenging dataset as each class only has 500 images. Our approach is generally applicable to not only vision tasks, but also natural language processing tasks. To show this, we further conduct experiments on Shakespeare, which is a text dataset built from Shakespeare Dialogues [35], and the task is next-character prediction. More details about these datasets can be found in the supplementary material.

**Data Partition.** In our experiment, we follow [19] and consider both cases of IID and non-IID data partition, in order to comprehensively showcase the performance. For IID partition, we uniformly sample a same number of data for each client. For non-IID case, we assume *label shifts* and distribute a subset of classes for each client. Specifically, the number of classes is set to 3 for Fashion-MNIST and CIFAR10, 30 for CIFAR100. For Shakespeare, we follow the partition method in [24]. We refer the underlying distribution of raw data (each speaking role) as the non-IID partition and each data point is equally likely to be sampled in the IID partition.

**Model Architecture.** Our approach can seamlessly work with both convolutional and fully-connected layers. In this paper, we evaluate it across several different architectures. Specifically, we adapt the standard ResNet-18 [21] for CIFAR10 and CIFAR100. For Fashion-MNIST, we adopt a simple 4-layer CNN. For next-character prediction, we apply our approach upon a 1-layer RNN. Further details can be found in the supplementary file.

**Client Partition.** For all experiments, we follow previous conventions [15] and assume totally 100 clients with 10% of them being active for each communication round. To model system heterogeneity,

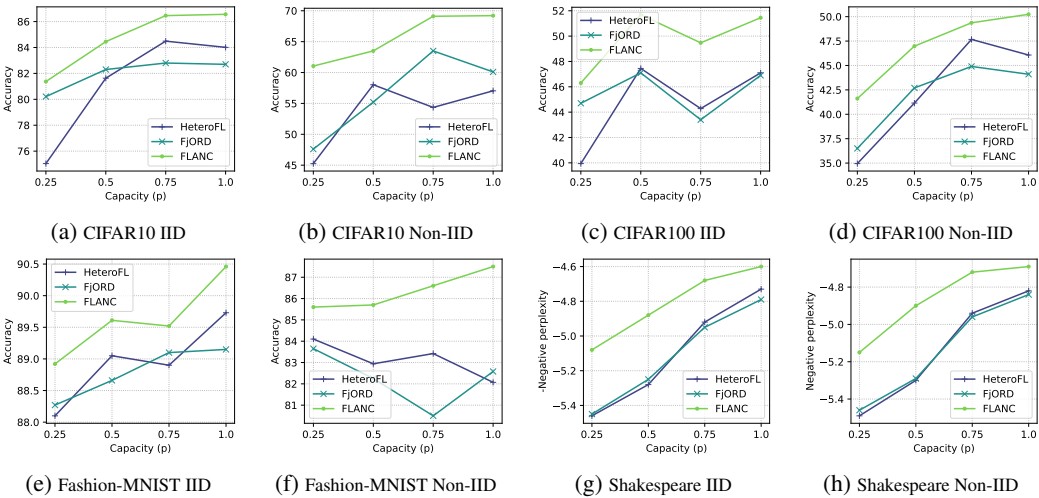

| (a) CIFAR10 IID | (b) CIFAR10 Non-IID | (c) CIFAR100 IID | (d) CIFAR100 Non-IID |

| (e) Fashion-MNIST IID | (f) Fashion-MNIST Non-IID | (g) Shakespeare IID | (h) Shakespeare Non-IID |

Figure 3: Test results for *static* system-heterogeneous setting. Top-1 ($\uparrow$) accuracy and negative perplexity ($\uparrow$) are used for image classification and next-character prediction tasks respectively.

we construct 4 different complexity levels with $\mathcal{P} = \{0.25, 0.5, 0.75, 1.0\}$. To fully demonstrate the effectiveness of our approach, we consider both the *static* and *dynamic* capacity distribution. For *static* setting, we uniformly assign a capacity level for each client, and then client keeps this budget throughout the training. For *dynamic* setting, we allow each client to randomly increase/decrease its capacity at every communication round. This is also a realistic scenario as devices' capacity can vary drastically due to other irrelevant system processes or CPU frequency changes led by the battery status.

**Baseline Methods and Implementation Details.** We compare our method with two state-of-the-art system-heterogeneous solutions – HeteroFL[19] and FjORD[20]. Both approaches are similar in design and are based on model pruning. Other generic FL algorithms [15, 14, 36] are inapplicable with the presence of heterogeneous devices due to broken assumptions. Since the official code for FjORD [20] is absent, we re-implement this method following the author's descriptions and use the default parameter settings including the uniform dropout rate. For hyper-parameter tuning, we split a subset of 10% training examples as the validation set. After selecting the parameters, validation data are merged back to the training set, and then we retrain the model for final performance evaluation. For our method, the selection of $\lambda$, $R_1$ and $R_2$ depends on architecture and tasks. We implement the proposed approach using PyTorch 1.8.2 on Nvidia A5000 GPUs. Detailed descriptions of hyper-parameters and training can be found in the Appendix.

### 3.1 Main Performance Evaluation

Here we compare our approach with HeteroFL [19] and FjORD[20] on standard benchmarks. All results with *static* setup are shown in Figure 3. Results with *dynamic* setting are reported in Table 1.

In both system-heterogeneous settings, our approach consistently outperforms both HeteroFL and FjORD on all entries. The improvements is further magnified when dealing with heterogeneous data. Specifically, we found the performance of HeteroFL [19] and FjORD [20] very close in many cases. This is consistent with our expectation as both methods are based on parameter pruning, even if FjORD [20] in addition allows large devices to randomly switch to a smaller sub-model through ordered dropout. However, we do not observe consistent improvements with this dropout technique, and in some cases it further harms the performance of the larger models. And both HeteroFL and FLANC are compatible with this technique. Furthermore, when switching from IID to non-IID data partition, both approaches suffer from significant performance drop. For example, on CIFAR10, the overall performance of both FjORD and HeteroFL drops to 60% even with their largest model. This is mainly because pruning restricts parameters to have a limited access to the full range of data and thus may not transfer to unseen distributions. In contrast, our approach allows every parameter to leverage a full range of knowledge through the shared basis, and hence is more robust to data heterogeneity. As a result, our FLANC achieves around 13% and 9% performance gain compared to HeteroFL and FjORD respectively.

Table 1: Experimental results with *dynamic* system-heterogeneity. For image classification, we use the standard Top-1 accuracy (%) as the metric. Perplexity is used as the metric on Shakespeare. The 'M' and 'K' denote $\times 10^6$ and $\times 10^3$ respectively.

| Width | Top-1↑ / Perplexity↓: IID | | | Top-1↑ / Perplexity↓: Non-IID | | | Communication Cost | |
| | HeteroFL | FjORD | FLANC | HeteroFL | FjORD | FLANC | p-width | FLANC |
|---|---|---|---|---|---|---|---|---|
| | | | CIFAR10 with ResNet-18 | | | | | |
| ×0.25 | 81.9 | **85.1** | **85.1** | 59.9 | 51.7 | **63.7** | 0.7M | **0.5M** |
| ×0.50 | 85.0 | 86.4 | **87.9** | 54.6 | 58.4 | **69.7** | 2.8M | **1.2M** |
| ×0.75 | 87.4 | 86.8 | **88.8** | 57.5 | 66.2 | **70.6** | 6.3M | **2.4M** |
| ×1.0 | 88.1 | 86.7 | **89.5** | 56.7 | 65.7 | **73.0** | 11.1M | **4.0M** |
| Avg | 85.6 | 86.3 | **87.8** | 56.2 | 60.5 | **69.3** | 5.2M | **2.0M** |
| | | | CIFAR100 with ResNet-18 | | | | | |
| ×0.25 | 49.2 | 52.5 | **54.2** | 44.2 | 46.7 | **48.3** | 0.7M | **0.5M** |
| ×0.50 | 56.5 | 54.9 | **61.8** | 50.6 | 47.5 | **54.7** | 2.8M | **1.2M** |
| ×0.75 | 61.3 | 55.8 | **64.2** | 52.1 | 49.2 | **56.5** | 6.3M | **2.4M** |
| ×1.00 | 63.6 | 55.8 | **65.5** | 52.7 | 48.8 | **57.1** | 11.1M | **4.0M** |
| Avg | 57.7 | 54.8 | **61.4** | 49.9 | 48.1 | **54.2** | 5.2M | **2.0M** |
| | | | Fashion-MNIST with CNN | | | | | |
| ×0.25 | 89.4 | 88.3 | **90.5** | 81.3 | 81.6 | **82.0** | 43K | **33K** |
| ×0.50 | 90.5 | 88.6 | **91.1** | 83.7 | 83.1 | **85.7** | 106K | **63K** |
| ×0.75 | 90.5 | 89.1 | **91.2** | 83.1 | 85.5 | **86.0** | 186K | **98K** |
| ×1.00 | 91.1 | 89.1 | **91.4** | 84.6 | 83.7 | **86.8** | 285K | **136K** |
| Avg | 90.4 | 88.8 | **91.1** | 83.2 | 83.5 | **85.1** | 155K | **83K** |
| | | | Shakespeare with RNN | | | | | |
| ×0.25 | 5.54 | 5.54 | **5.17** | 5.55 | 5.54 | **5.21** | 49K | **48K** |
| ×0.50 | 5.07 | 5.06 | **4.74** | 5.09 | 5.05 | **4.80** | 214K | **187K** |
| ×0.75 | 4.86 | 4.87 | **4.64** | 4.92 | 4.94 | **4.65** | 443K | **417K** |
| ×1.00 | 4.64 | 4.67 | **4.52** | 4.69 | 4.73 | **4.54** | 787K | **740K** |
| Avg | 5.03 | 5.04 | **4.77** | 5.06 | 5.07 | **4.80** | 373K | **348K** |

In terms of efficiency, all three methods are able to straightforwardly reduce the amount of data to transfer and the cost of computation by switching to smaller $p$-width models. However, benefiting from the efficient tensor decomposition format, our approach can further reduce the model size, resulting in the best communication efficiency.

## 3.2 Robustness with Imbalanced Device Distribution

In practice, there is usually no guarantee that different types of devices will be equally distributed. Here we conduct experiments to compare the robustness towards imbalanced device distributions. To make things more challenging, we also assume there is a large capacity gap between clients. Specifically, we consider two types of clients A and B, where A can run the full model with $p$=1 and B has a limited support of $p$=0.25 . We consider two scenarios where A accounts for 25% and 75% of total devices respectively.

The results are illustrated in Figure 4. It is noted that, with a limited number of available devices of type A, HeteroFL and FjORD cannot effectively improve the accuracy by increasing the capacity, but instead suffer from performance degradation. This is not surprising as the majority of parameters (>90%) is under-trained with only $1/4$ portion of all data available. In contrast, our approach is still able to obtain performance gain with additional parameters. This proves that our approach indeed is more favorable for addressing system heterogeneity issues for FL. On the other hand, when the majority of devices can run the original model, the performance gap between ours and HeteroFL shrinks as expected, since in this case most of the parameters can directly access all data. In contrast,

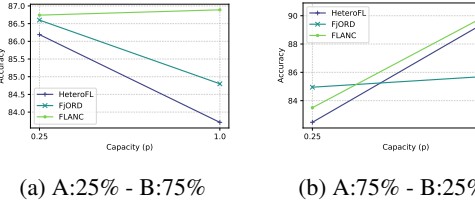

(a) A:25% - B:75%          (b) A:75% - B:25%

Figure 4: Comparison with HeteroFL and FjORD for imbalanced client distributions. Results are reported on CIFAR10 under *dynamic* system heterogeneity and IID data partition. (a) Majority participants are indigent devices of type B. (b) Majority clients are high-end devices of type A.

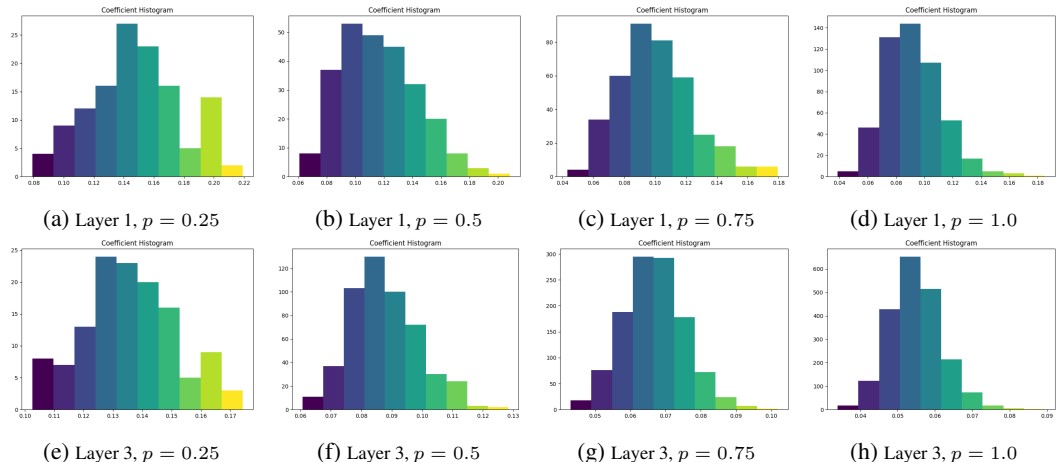

(a) Layer 1, $p = 0.25$    (b) Layer 1, $p = 0.5$    (c) Layer 1, $p = 0.75$    (d) Layer 1, $p = 1.0$

(e) Layer 3, $p = 0.25$    (f) Layer 3, $p = 0.5$    (g) Layer 3, $p = 0.75$    (h) Layer 3, $p = 1.0$

Figure 5: Statistics of mean coefficient value. Results for the first convolution in the corresponding residual layer are reported.

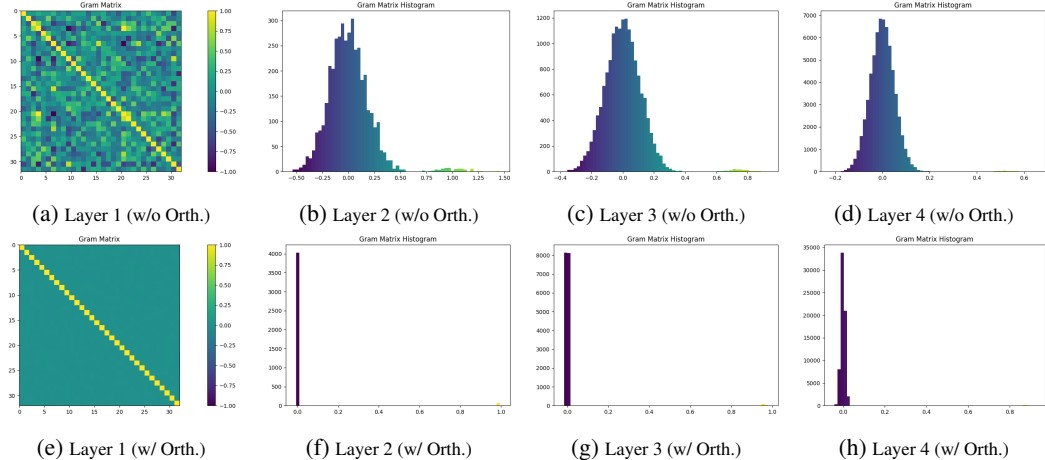

(a) Layer 1 (w/o Orth.)    (b) Layer 2 (w/o Orth.)    (c) Layer 3 (w/o Orth.)    (d) Layer 4 (w/o Orth.)

(e) Layer 1 (w/ Orth.)    (f) Layer 2 (w/ Orth.)    (g) Layer 3 (w/ Orth.)    (h) Layer 4 (w/ Orth.)

Figure 6: Visualization of the Gram matrix (of every first convolution in each residual layer) when w/ or w/o orthogonal regularization. Gram matrices for Layer 2-4 are shown as histograms because they are too large to clearly visualize.

FjORD squeezes its performance into some intermediate points due to dropout. While it improves the performance of the smaller model, the benefits from larger capacity become marginal.

### 3.3 Knowledge Sharing

Here we study the effectiveness of FLANC in terms of knowledge sharing. One can imagine that if all values in a set of coefficients (a column of $U_p$) are close to zero, the proposed neural composition will degenerate to pruning. To show that this is not the case, we report the statistics of the mean absolute value for each column in the coefficient matrices $U_p$. As shown in Figure 5, most values are not small and zero-centered, indicating knowledge stored in the basis can be well-shared across models.

### 3.4 Ablation Study

**Orthogonality.** Orthogonal regularizer reduces the redundancy and dependency within neural bases and improves the expressiveness of neural composition. To demonstrate its effectiveness, we remove the orthogonality regularization and compare the performance on CIFAR10. As shown in Figure 7a, one can observe that our regularizer constantly improves performance across all model capacities. We further visualize the Gram matrix of the learned basis in Figure 6. It can be seen that the Gram matrices of our approach are very close to the diagonal matrices, demonstrating the basis is indeed

orthogonal. It is worth noting that such property cannot be learned naturally with purely classification loss.

**Basis Size** $R_1$**.** In our implementation, we set the basis size $R_1$ as the common divisor of the input channels over all models. Thus, we can compose all networks with one unified basis. To study our model's sensitivity to the choice of $R_1$, we set it to 25%, 50%, 100% of the smallest incoming channel, respectively. As shown in Figure 7b, using fine-grained basis can slightly improves the accuracy. The performance peaks at 50% of the smallest incoming channel.

**Number of Bases** $R_2$**.** Increasing the number of bases inherently enlarges the dimensionality of the subspace and may improve performance

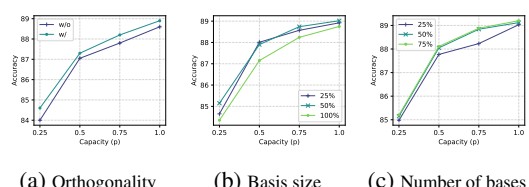

(a) Orthogonality     (b) Basis size     (c) Number of bases

Figure 7: Ablation study of FLANC. Results are reported on CIFAR100 with IID data partition and dynamic system heterogeneity.

by providing better representational capacity. We study its effect and consider $R_2$ to be 25%, 50% and 75% width of the original model, respectively. As shown in Figure 7c Results suggest that a larger number of bases can improve the performance but the benefit becomes marginal when further increasing from 50% to 75%. In addition, using more bases also linearly increases the model size. For our architectures, we select $R_2$ with the best trade-off between the performance and the number of parameters.

## 4 Related Works

**Data Heterogeneity in FL.** In the *de facto* FL algorithm (FedAvg [15]), clients conduct local training with the received parameters from a central server. Then the server collects all the locally trained models and aggregates them into a new global model. The design of FedAvg [15] relies on the assumption that data is uniformly distributed across clients. However, the assumption does not hold in real-world scenarios where the underlying data distribution is unknown and very likely non-IID. The non-IID distribution of clients' local data is one notorious trap in federated learning and results in the non-negligible client drift issue, which can jeopardize the convergence rate and model performance when the data similarity decreases [37]. Tackling the statistical data heterogeneity attracts extensive attention in recent years. Pioneer works, like FedProx [38], introduce additional regularizers on the local training objective to prevent the local models from diverging due to the non-IID data distribution. Later on, several works propose inter-client variance reduction techniques [36, 16, 39] by amending the client drift issue with the predicted model updating direction. Personalized FL [40, 41, 42, 43] is another strategy to allow a model to better fit the local data distribution on a specific client. One can fine-tune the global model on the local data [44], perform MAML-based personalized approaches [40, 41], or achieve the personalization by local batch normalization layers [14]. Our proposed method is perpendicular to the above studies and potentially can be combined with them for further improvement.

**System Heterogeneity in FL.** In contrast to the substantial effort devoted to data heterogeneity, addressing FL system heterogeneity is largely under-explored. The majority of existing methods only focus on reducing communication cost, so that the data transfer cost can be affordable by those edge devices with constrained bandwidth and energy consumption. For example, several advanced FL optimizers [16, 36, 45, 46] have been proposed to improve the convergence rate, and thus reduce data transfer cost by decreasing the number of required communication rounds. Other methods [47, 48, 49, 50] improve communication efficiency by combining quantization [47, 49] and sparse training [48, 51] techniques. For example, LotteryFL [43] and FedMask [42] learn personalized sparse sub-networks to achieve high communication efficiency. However, all aforementioned approaches still assume a uniform processing capacity among all participants, which cannot be guaranteed for real-world scenarios. One promising way to remove this assumption is to leverage knowledge distillation (KD) for learning the central model instead of conventional coordinate-wise averaging. While such methods [52, 53, 54, 55, 56] allow different local models, applying KD often requires additional access to some common public datasets to alleviate the difficulty in knowledge transmission [57], which is impractical in FL. For the most recent methods, Split-Mix [58] explores model ensemble by re-mixing universally-budget-compatible sub-networks at inference time, where its effectiveness is

potentially constrained by the insufficient capacity of individual sub-models. FedHM [59] explicitly factorizes global models into low-rank sub-models to accommodate client capacity, which has shown to be effective in reducing computation and communication costs. However, the SVD scheme preserves feature width before and after decomposition and thus are lack the ability to scale (peak) memory footprint and may harm the performance by introducing additional factorization error. Our method is most related to HeteroFL [19] and FjORD [20], which allow federated training of local models in different widths adaptively "pruned" from a global model. Nevertheless, as discussed before, they cannot fundamentally resolve the challenges and still suffer from degraded performance similar to many conventional FL methods.

**Compression Techniques.** Both our approach and previous heterogeneous-system solutions are related to model compression techniques [34, 30, 29, 60, 61, 62], a classic research field to reduce the run-time storage and the latency of deep neural networks. Some of the most representative techniques include quantization [62, 63], neural pruning [60, 61], and tensor decomposition [34, 30, 29]. In the context of FL, these techniques are first studied and demonstrated to be effective in reducing the communication cost [47, 48, 49, 50]. To handle system heterogeneity, current approaches [19, 20] extend model pruning techniques to extract sub-models to fulfill the resource requirements. However, as discussed, pruning simply drops model parameters, preventing the dropped ones from leveraging complete knowledge scattered across various devices. On the other hand, our method can be connected to tensor decomposition, where we express layers as a unified *neural basis* representing shared knowledge and capacity-specific coefficients for linear composition. In this way, it not only allows resource-adaptive model training through composition, but also naturally ensures the complete knowledge effectively propagates to models at various complexities.

**Factorization-based FL Methods.** In this work, the low-rank nature of deep models provides a foundation for using a unified neural basis to express knowledge and compose heterogeneous models. Compared to existing factorization-based approaches [59, 64], our approach does not perform explicit low-rank decomposition but instead only uses this notion conceptually, i.e., FLANC directly learns the basis and coefficients from scratch rather than factorizes them from existing kernels.

This brings several advantages compared to previous approaches performing real low-rank decomposition: **(1)** FLANC avoids approximation errors of factorization, which will potentially harm the performance. Meanwhile, it can be applied to every layer for better complexity scaling. In contrast, SVD based factorization inevitably introduces an approximation error that will be accumulated and propagated to subsequent layers. This limits their applicability to the last several layers and reduces practicality for real-world deployment, as early layers can also become computational and memory bottlenecks, for example, when dealing with large images. **(2)** FLANC can handle system heterogeneity by constructing client models with different widths, whereas factorization methods cannot achieve the same. Explicit factorization requires the resulting tensors to have aligned input and output dimensions (e.g., $m \times r$ and $r \times n$) with the original kernel (e.g., $m \times n$), which cannot change run-time width to effectively adjust memory consumption. FedDLR [64] and FedPara [50] further assume a uniform processing capacity over devices, making them infeasible for dealing with system heterogeneity.

## 5 Conclusion

The assumption of uniform processing capacity in conventional federated learning contradicts the varying computation resources in practice. As a result, these methods suffer from an inevitable dilemma between model complexity and data accessibility. To overcome this problem, we propose a simple yet effective mechanism, termed All-In-One Neural Composition to systematically support resource-adaptive federated learning. Unlike previous methods, our method enables flexible constructions of models in different complexities, and allows unhindered access to the full range of knowledge scattered across clients. Comprehensive experimental results on vision and language tasks demonstrate the effectiveness of our method, manifesting consistent performance gain across different setups.

## Acknowledgments and Disclosure of Funding

This work was supported by an ARO grant W911NF-21-1-0135.

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
