# Supplementary File:
# Resource-Adaptive Federated Learning with All-In-One Neural Composition

**Yiqun Mei  Pengfei Guo  Mo Zhou  Vishal M. Patel**
Johns Hopkins University
{ymei7,pguo4,mzhou32,vpatel36}@jhu.edu

## 1 Experiment Details

### 1.1 Dataset Details

For image classification, we consider three common datasets[1]: Fashion-MNIST [1], CIFAR10, and CIFAR100 [2]. Fashion-MNIST consists of 60K training images and 10K testing images. Each example is a $28 \times 28$ gray-scale image. It consists of 10 classes of fashion products, and usually servers as a drop-in replacement for the original MNIST [3]. Both CIFAR10 and CIFAR100 datasets contain a training set of 50K examples and a test set of 10K examples. Each example is a $32 \times 32$ RGB image associated with a label from 10 and 100 classes, respectively. Shakespeare provides 422,615 samples with a sequence length of 80. We follow [4] to split the dataset into 90% for training and 10% for testing.

### 1.2 Model Architectures

We train the standard ResNet-18 [5] on CIFAR10 and CIFAR100 datasets. For each capacity $p \in \mathcal{P}$, we follow [6] and maintain independent batch normalization layers for each capacity group. We perform model pruning described by HeteroFL [7] and FjORD[6], and the proposed neural composition for all layers except the input layer and the linear classifier. The basis size $R_1$ is set to 50% of the smallest incoming channel for all layers (i.e. 12.5% of the full model). The number of bases $R_2$ is set to 25% of the output channel of the full model. Standard data augmentation methods, i.e. random cropping and horizontal flipping, are used during training.

For Fashion-MNIST, we construct a simple 4-layer CNN with 3 convolutional layers and a linear classification head. Each convolution has the kernel size of $3 \times 3$ and 64, 128, 128 hidden channels, respectively. Model pruning and neural composition are performed for all convolutional layers. Similar to ResNet-18, We set $R_1$ to $1/2$ of the smallest input channel and $R_2$ to $1/4$ of the full output channel. No data augmentation is used for this experiment.

For Shakespeare, we construct a simple 1-layer RNN and set both the hidden channel size and embedding size to 512. We set the basis size $R_1$ to 32 and the number of bases $R_2$ to 30 for the RNN cell. No data augmentation is used.

### 1.3 Training Hyperparameters

Detailed hyperparameters are summarized in Table 1. For all experiments, we use fixed seed and PyTorch deterministic algorithms[2] for fair comparisons. Error bars are also studied in Figure 3.

---

[1]Fashion-MNIST is under the MIT License. Licenses for the rest datasets are unspecified.
[2]https://pytorch.org/docs/stable/notes/randomness.html

36th Conference on Neural Information Processing Systems (NeurIPS 2022).

Table 1: Hyperparameters used in our experiments

| Dataset | Data partition | Fashion-MNIST | CIFAR10 | CIFAR100 | Shakespeare |
|---|---|---|---|---|---|
| Model | | CNN | ResNet-18 | ResNet-18 | RNN |
| Optimizer | | | SGD | | |
| Momentum | | | 0.9 | | |
| Weight decay | | | 5e-4 | | |
| Batch size | | 32 | 32 | 32 | 1000 |
| Learning rate | | 0.01 | 0.1 | 0.1 | 0.1 |
| Communication rounds | IID | 200 | 500 | 500 | 80 |
| | Non-IID | 200 | 800 | 800 | 80 |
| Local epochs | IID | 1 | 2 | 2 | 1 |
| | Non-IID | | 1 | | |
| LR decay (0.1) | IID | [100] | [250, 375] | [250, 375] | [20,40] |
| | Non-IID | [100] | [400, 600] | [400, 600] | [20,40] |
| $\lambda$ (for FLANC) | | 0.1 | 10 | 10 | 10 |

## 2  Communication Complexity

In the paper, we claim that our method is communication efficient. Here we provide a detailed analysis. Following the notation defined in Section 2 in the main paper, our method express a standard $p$-width convolutional kernel of shape $k^2 \times Sp \times Tp$ as linear combinations of a neural basis $V_{\text{share}}$ of shape $k^2 \times R_1 \times R_2$ and capacity specific coefficients $U_p$ of shape $R_2 \times Sp/R_1 \times Tp$. Therefore, compared to the standard $p$-width model, the number of parameters required to be transferred is

$$\frac{k^2 R_1 R_2 + (R_2/R_1)STp^2}{k^2 STp^2} = \frac{R_1 R_2}{STp^2} + \frac{R_2}{k^2 R_1} \leq 1$$

by properly setting $R_1$ and $R_2$ (e.g., $R_1 = 1/2Sp$ and $R_2 = Tp$ when $T = 2S$). This leads an on-par or better communication efficiency compared to the pruning-based solutions in our experiments.

## 3  Computational Complexity

In our paper, we claim that our method is also highly efficient in terms of computational complexity. Here we discuss its computational implication.

In the practical implementation, the proposed All-In-One Neural Composition can be realized in two equivalent ways: (1) explicitly creating a weight $W$ before every forward pass by multiplying $V_{\text{share}}$ and coefficients $U$, and convolving over the input using the resulting $W$; (2) maintaining the factorized tensor format by performing two consecutive convolutions with $V_{\text{share}}$ and $U$. The second strategy is based on the linearity of convolutions so that one can reorder the operations, *i.e.*, $(V_{\text{share}} \cdot U)^T X = U^T (V_{\text{share}}^T \cdot X)$ for an input $X$. In the following, we show that the first strategy brings negligible computational overhead compared to the standard convolution, and the second strategy further improves the efficiency.

Given a batch of $b$ input features with spatial size $q \times q$ and input channel number $c$, performing a standard convolution with kernel size $k \times k$ and output channel number $c$ requires $q^2 k^2 c^2 b$ multiply-add float operations.

For the first strategy, the additional complexity comes from composing the weight for a batch of data. This extra cost from matrix multiplication of $V_{\text{share}} \cdot U$ is $k^2 R_1 \cdot R_2 \cdot (c/R_1)c = k^2 c^2 R_2$. Compared to the standard convolution, the additional cost of this strategy is

$$\frac{k^2 c^2 R_2}{q^2 k^2 c^2 b} = \frac{R_2}{q^2 b} \ll 1 \tag{1}$$

which is negligible when $R_2 \ll q^2 b$.

The above procedure is also equivalent to conduct two consecutive convolutions with $V_{\text{share}}$ and $U$. Specifically, the input $X$ is first split into $c/R_1$ chunks of $R_1$ channels and convolve with the $V_{\text{share}}$. Then the final output is obtained by linear summation of the splits using $U$, which is equivalent to performing a $1 \times 1$ convolution. Convolving over $c/R_1$ splits using $V_{\text{share}}$ requires $q^2 k^2 c R_2$ multiply-add operations. And the complexity for the linear summation is $q^2 c^2 (R_2/R_1)$. Compared

Table 2: Comparison of computational efficiency in terms of MACs. HeteroFL [7] and FjORD [6] have the same MACs as the $p$-width model. The factorized tensor format used in our method achieves better overall efficiency.

| Models | CNN | | | | ResNet-18 | | | | RNN | | | |
|---|---|---|---|---|---|---|---|---|---|---|---|---|
| | $\times 0.25$ | $\times 0.50$ | $\times 0.75$ | $\times 1.00$ | $\times 0.25$ | $\times 0.50$ | $\times 0.75$ | $\times 1.00$ | $\times 0.25$ | $\times 0.50$ | $\times 0.75$ | $\times 1.00$ |
| $p$-width | **17M** | 37M | 60M | 88M | **45M** | 153M | 325M | 557M | 5M | 17M | 38M | 66M |
| Ours | 20M | **26M** | **33M** | **41M** | 55M | **112M** | **186M** | **275M** | **5M** | **16M** | **34M** | **59M** |

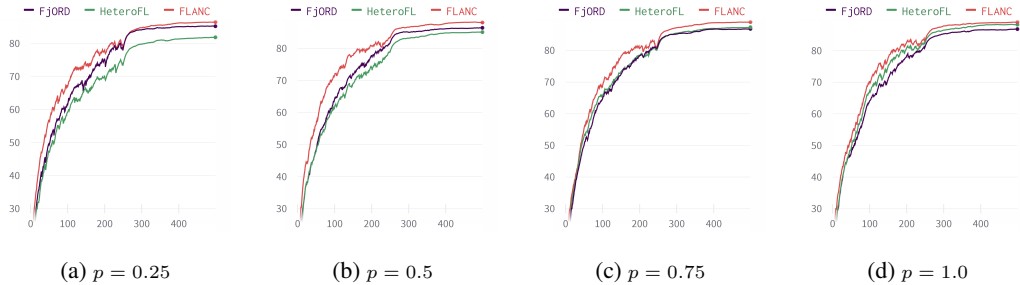

(a) $p = 0.25$       (b) $p = 0.5$       (c) $p = 0.75$       (d) $p = 1.0$

Figure 1: Test accuracy curves w.r.t communication rounds.

with the standard convolution, the total computational cost required by the factorized tensor format is

$$\frac{q^2 k^2 c R_2 + q^2 c^2 (R_2/R_1)}{q^2 k^2 c^2} = R_2/c + R_2/(k^2 R_1) < 1 \tag{2}$$

under our experiment setting (*e.g.*, kernel size $k = 3$, $R_1 = 0.125c$, and $R_2 = 0.25c$ for image classification). In Table 2, we report the computational cost and compare it with the original $p$-width network used by HeteroFL and FjORD. The results show our approach is indeed more efficient.

## 4  Additional Experiments

### 4.1  Convergence Speed

In Figure 1, we report test accuracy curves w.r.t training rounds on CIFAR10 and compare with HeteroFL [7] and FjORD [6]. The results show the proposed FLANC has on-par or faster convergence speed and achieves better accuracy than baselines.

### 4.2  Complexity Levels ($p$)

Here we study the effect of device complexity levels. We model more fine-grind capacity groups, which cover a wide range of device capacities, ranging from 1.5% to 100% in terms of computational and communication cost and 12.5% to 100% in terms of memory. And only one device from each capacity group is allowed to join for a training round so that each joined device has a unique complexity level. We report the results in Figure 2. One can see that our approaches can still stably outperforms baselines. However, we speculate it would be challenging when the number of devices of distinct complexity levels becomes large, as the specific coefficient (only shared among one device) will be biased towards its own data. To this end, the introduced pre-clustering strategy, which quantizes complexity levels, is important to stabilize the training.

### 4.3  Effects of Randomness

We conduct five experiments with different random seeds on CIFAR10 and report the results in Figure 3. One can see that the FLANC can stably outperform existing baselines, and the performance improvement is beyond the error range.

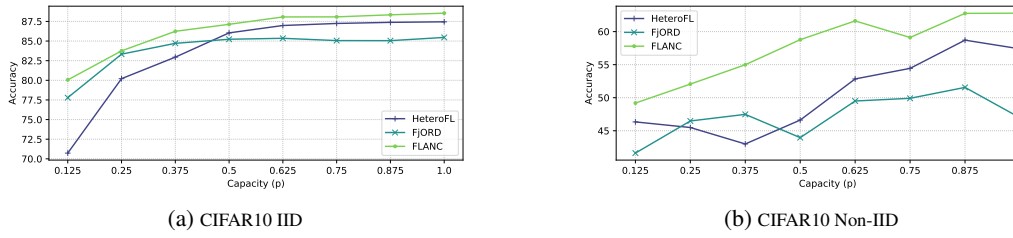

(a) CIFAR10 IID               (b) CIFAR10 Non-IID

Figure 2: blueResults of more complexity levels ($p$).

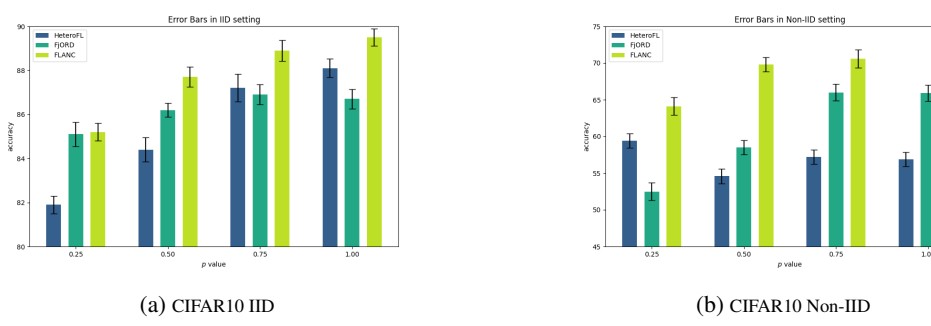

(a) CIFAR10 IID               (b) CIFAR10 Non-IID

Figure 3: Statistics of test accuracy on CIFAR10 with dynamic system heterogeneity. The error bars represent 95% confidence intervals from 5 repeats.

## 5  Real to Abstract Capacity Mapping

Due to the lack of mature software support, we did not benchmark the performance on real devices but instead abstract the resource constraints as model width for simplicity and demonstration purposes. While it is interesting to connect our abstraction to real device capacity, constructing such accurate mapping requires solving many engineering challenges and we leave it for future work. However, it is possible to roughly map real GPUs/devices to a complexity level $p$ based on the memory capacity. We report some examples of popular GPUs and mobile devices in Table 3. We would also like to mention some public benchmarking results (e.g., AI Benchmarks) for real deployment.

## 6  Broader Impact

Federated learning provides a decentralized solution for large-scale machine learning. Compared to centralized systems, FL alleviates privacy concerns and is more economic, as it makes it possible to leverage computational power from billions of edge devices worldwide. This removes the need of constructing huge data centers and reduces the carbon footprint of transferring large amounts of raw data. Nevertheless, standard FL algorithms [8, 9, 10, 11, 12, 13] suffer from system heterogeneity and the potential exclusion of low-end devices may lead to biased predictions and fairness issues, because device preferences can be connected to demographic information of users. Compared with existing solutions [6, 7], our approach is not only more economic by allowing all devices to participate, but also improves fairness and alleviates prediction bias by enabling every client, no matter how much their processing capacities are, to make unhindered knowledge contribution to the entire system.

Although FL is favorable for many real-world applications, it still risks malicious usage if the users are unaware of their resources and private data being used by service providers. To prevent such cases, consents must be obtained from end users when deploying FL in the wild. In addition, recent research shows that, even without raw data transferring, information disclosure is still possible with gradient leakage attacks [14, 15, 16], emphasizing the necessity of advanced defense techniques.

## 7  Limitations and Future work

In this work, we propose a new algorithm to achieve federated training with unaligned client capacity. We demonstrate the superiority of our approach on extensive simulated FL setups. However, due to the lack of mature software support, we did not benchmark the performance on real devices. In

Table 3: Examples of real devices mapping to abstract capacity levels by supposing running a full model with 10GB memory requirement.

| Device type | RTX 3090 | Nvidia V100 | RTX 3080 | Radeon RX 580 | Arc A380 | GTX 1060 | GTX 970 | Iphone 13 | Quadro K4000 |
|---|---|---|---|---|---|---|---|---|---|
| Memory (G) | 24 | 16 | 10 | 8 | 6 | 6 | 4 | 4 | 3 |
| Ratio ($p$) | 1.0 | 1.0 | 1.0 | 0.75 | 0.5 | 0.5 | 0.25 | 0.25 | 0.25 |

addition, we also abstract the resource capacity as model width for simplicity and demonstration purposes, indicating a lookup table that maps hardware to the width of the network is required for practical applications. We leave these engineering challenges raised by real-world deployments for future work.

In our experiments, we demonstrate that our approach can enable federated training over at least eight capacity groups, which covers a wide range of devices in practice (from 1.5% to 100% in terms of computation and communication overheads and 12.5% to 100% in terms of memory). However, we acknowledge that it will be challenging to handle many capacity groups at very fine-grained levels, especially when the group size is further limited. In this case, the learned coefficients will be potentially biased towards the corresponding particular group of devices. The provided pre-clustering strategy is required for quantizing complexity levels.

Without defense techniques, privacy leakage is a common challenge for almost all FL methods. Different from existing algorithms which will disclose global statistics over the entire user group, our method will reveal statistics with respect to the specific capacity groups. However, we do believe our approach offers better privacy protection by design, as information leakage cannot happen without the simultaneous leakage of both neural basis and coefficients. Advanced techniques are also required to combine and decode the information. Further combining our approach with advanced privacy-preserving techniques, such as differential privacy and homomorphic encryption, will be an interesting direction for future exploration.

When the resource constraint changes within a single communication round, the device may not necessarily be able to support the current model and the server has to either exclude or drop those "stragglers" in case of timeout. One intuitive way to tackle this situation is to simply send all coefficients to the clients and let the clients switch to a suitable model upon its resource allocation, but meanwhile, this induces higher communication costs. It might also be interesting to improve the straggler-resilience of our approach. We leave this extension for future work.

In addition, similar to previous studies [6, 7], we apply a uniform rescale ratio $p$ to all layers for simplicity. Nevertheless, our approach is generally applicable to support varying $p$ for individual layers. This makes searching for the best architectures (in terms of width) possible and is intriguing to explore in the future.

Finally, this paper focuses on the classical federated learning setting which aims to output a global model that generalizes well on all clients' data. Since our approach is generic, it can be potentially extended to address personalized federated learning with heterogeneous devices. We will study this extension in future work.