# OpenReview forum: "Resource-Adaptive Federated Learning with All-In-One Neural Composition"
_NeurIPS.cc/2022/Conference — NeurIPS 2022 Accept_

### Official Review · Reviewer_dZNT · 2022-07-09

**Rating:** 7
**Confidence:** 2
**Soundness:** 3 good
**Presentation:** 3 good
**Contribution:** 3 good

**Summary:**

This paper presents FLANC, a model decomposition method for the heterogeneous devices in Federated Learning (FL). The basic idea is to change the model size/complexity by varying the ratio of channels.

**Questions:**

How does FLANC perform with regards to other FL aspects such as training rounds and communication overheads?

**Strengths And Weaknesses:**

Strengths
1.Better supporting of heterogeneous devices (e.g., IoT and mobiles) is an important research topic in FL.
2.FLANC is generally applicable to other models and datasets.
3.FLANC shows promising results compared to HeteroFL and FjORD.
4.This paper is well-written and easy to follow.

Weaknesses
1.Only the metric of accuracy is evaluated. It is better to consider other important aspects of FL, e.g., communication overheads.
2.Evaluation results are from one-run experiment only, which is less convincing.

---

> ### Author Response · Authors · 2022-08-01
> **Response to Reviewer dZNT**
>
> Thanks very much for your valuable comments. We are glad that you recognize the significance of the topic, broad applicability and promising results of the method, and good writing of this paper. We address the concerns in the following text and have changed the manuscript accordingly.
>
> **wrt communication overhead.** The communication overhead is reported as "model size'' in the last two columns of Table 1. We have changed the column name to "communication cost'' to avoid confusion. One can see that our approach can achieve better overall efficiency compared to previous solutions. We also add a detailed complexity analysis in the supplementary file.
>
> **wrt training rounds.** We included a discussion on training rounds in the supplementary material. As shown in Figure 1, our approach has a faster/comparable convergence speed and achieves better accuracy compared to previous solutions.
>
> **wrt experimental results with error bar.** As explained in Checklist 3(c), it is computationally expensive to report error bars for all experiments given a limited time frame because our evaluations involve training 192 individual models. We do try to minimize random effects by fixing all seeds and using [PyTorch deterministic algorithms](https://pytorch.org/docs/stable/generated/torch.use_deterministic_algorithms.html#torch.use_deterministic_algorithms) for a fair comparison. However, we agree that the multi-run experiment will make our results more convincing. Therefore, we conduct 5 times of experiments with different random seeds on CIFAR10 and measure the variance. The corresponding results are reported in Figure 4 of the supplementary file, which shows our method can consistently outperform existing solutions.

---

### Official Review · Reviewer_83ZB · 2022-07-11

**Rating:** 6
**Confidence:** 2
**Soundness:** 3 good
**Presentation:** 3 good
**Contribution:** 3 good

**Summary:**

This paper presents the design of All-In-One Neural Composition, which can formulate networks at different complexities with one unified neural basis that's shared among clients. FL systems have suffered from the problem of system heterogeneity where the service providers would either sacrifice model capacity or data accessibility. This paper provides the FLANC framework that provides systematic support to train complexity adjustable neural networks and evaluated on a wide range of statistical data heterogeneity and system heterogeneity.

**Questions:**

1. In the evaluation section, the authors varied the $p$ to simulate heterogeneous systems. It's good to also include some real cases including the exact CPU/GPU that might result in the different $p$ (i.e. the limitations of computation power on each device to result in the limitation on model size).

**Limitations:**

Yes

**Strengths And Weaknesses:**

Strength:
1. Proposed a runtime method to construct the compressed model instead of applying after the training process compared to other low-rank techniques.
2. Conducted extensive experiments to show the effectiveness of the proposed method.

Weakness:
1. Didn't include error bars in the evaluation results. The results reported may not be statistically confident without any error bars. The authors could add randomness experiments on some small scale experiments.

---

> ### Author Response · Authors · 2022-08-01
> **Response to Reviewer 83ZB**
>
> We thank the reviewer for their helpful comments and valuable time spent reviewing our manuscript. We are glad that the reviewer acknowledges the contribution, effectiveness and extensive experiments of this paper. We address the concerns below and have also updated our manuscript accordingly.
>
> **wrt error bars.**  We agree that adding error bars will indeed make our quantitative evaluations more convincing. To this end, we conduct five experiments with different random seeds on CIFAR10 and report the results in Figure 4 of the supplementary file. These results show our method can stably outperform existing solutions, and the performance improvement is beyond the error range.
>
> **wrt real devices.** As discussed in the supplementary file (original version, lines 76-81), due to the lack of mature software support, we did not benchmark the performance on real devices. We instead abstract the resource constraints as model width for simplicity and demonstration purposes. Constructing an accurate mapping from hardware specifications to the ratio $p$ requires solving engineering challenges and we leave it for future work. However, it is possible to roughly map real devices to complexity level $p$ based on the GPU memory capacity. We include some examples  of popular GPU/device types in the following:
> |Device type  | Memory (G)  | ratio $p$  |
> |---|---|---|
> | RTX 3090        | 24  | 1.0 |
> | Nvidia V100     | 16 | 1.0 |
> |RTX 3080   |  10 | 1.0  |
> |Radeon RX 580, RTX 3070Ti, RTX 3060Ti, RTX 2070 |8  | 0.75  |
> |Arc A380, RTX 2060 | 6| 0.5|
> |GTX 1060 |6|0.5|
> |GTX 970, IPhone 13 | 4| 0.25|
> |Quadro K4000, IPhone 8 Plus |3|0.25|
> by supposing running a full model with 10GB memory requirement.  We have included this discussion in the supplementary file. We would also like to mention some public benchmarking results (e.g. https://ai-benchmark.com/ranking_deeplearning_detailed.html) for real deployment.

---

### Official Review · Reviewer_f2kP · 2022-07-20

**Rating:** 5
**Confidence:** 4
**Soundness:** 2 fair
**Presentation:** 3 good
**Contribution:** 2 fair

**Summary:**

Practical federated learning (FL) needs to handle heterogenous resource constraints among participants. Existing solutions mostly rely on pruning a global model in each client, which would cause uneven contributions to the global model weights among different clients. The paper proposes a mechanism that factorizes the global model weights into two low-rank tensors, where one of the tensors is unified across all the clients (called neural basis) and the other tensor varies according to clients’ resource. The paper also proposes a regularization term to encourage the basis vectors to be orthogonal with unit term. The evaluation shows the proposed mechanism outperforms two existing solutions, HeteroFL and FjORD.

**Questions:**

1. Can you provide more evidence to support the claim that the shared neural basis enables more knowledge sharing than pruning among clients? If the values in the coefficient matrix are small, isn’t it effectively similar to pruning?
2. How will the proposed method work if each device has a unique complexity level? Will the training still converge?


**Limitations:**

The paper does not provide meaningful discussions on its limitations, and it would be good to see the discussion on the following aspects:
1. How will the proposed mechanism work with fine-grained complexity levels?
2. What are the privacy implications of sharing the coefficient matrices?
3. What if the resource constraint changes within a communication round (e.g., other applications running)?


**Strengths And Weaknesses:**

Strengths
-------------
1. The paper proposes a reasonable solution to an important problem in FL.
2. The paper proposes a mechanism that generalizes pruning-based solutions to this problem.
3. The proposed regularization term is interesting.

Weaknesses
----------------
1. The paper misses important baselines in this area. Specifically, the paper should also compare its solution against LotteryFL (Li et al., 2020), FedHM (Yao et al., 2021), and FedMask (Li et al., 2021). These three solutions aim to address the same problem, and they report better accuracy than the paper.
2. The novelty is incremental. The idea of using low-rank factorization to address system heterogeneity has been proposed by FedHM (Yao et al., 2021). The general idea of using low-rank factorization for FL was proposed by FedDLR (Qiao et al., 2021) and FedPara [47]. While the detail of factorization might be different, it is unclear if the proposed factorization technique is better than other existing ones.
3. There is no analysis on computation and communication complexity. Similarly, there is no convergence analysis. The paper should provide an analysis of the proposed mechanism, and it would be much better to do the same against related solutions in this space.

---

> ### Author Response · Authors · 2022-08-02
> **Part 3: Response to Reviewer f2kP**
>
> **wrt suggestions on limitations.** Thanks for your helpful suggestions! We discuss the limitations on the mentioned aspects below and have added them to the manuscript accordingly.
>
> - In our experiments, we demonstrated that our approach can enable federated training over at least eight capacity groups, which covers a wide range of devices in practice (from 1.5\% to 100\% in terms of computational and communication overheads and 12.5\% to 100\% in terms of memory). However, we acknowledge that it will be challenging to handle many capacity groups at very fine-grained levels, especially when the group size is further limited. In this case, the learned coefficients will be potentially biased towards the corresponding particular group of devices. The provided pre-clustering strategy is required for quantizing complexity levels.
>
> - Without defense techniques, privacy leakage is known to be a common challenge for almost all FL methods. Different from existing algorithms which will disclose global statistics over the entire user group, our method will reveal statistics with respect to the specific capacity groups. However, we do believe our approach offers better privacy protection by design, as information leakage cannot happen without the simultaneous leakage of both neural basis and coefficients. Advanced techniques are also required to combine and decode the information. Further combining our approach with advanced privacy-preserving techniques, such as differential privacy and homomorphic encryption, will be an interesting direction for future exploration.
>
> - When the resource constraint changes within a single communication round, the device may not necessarily be able to support the current model and the server has to either exclude or drop those "stragglers" in case of timeout. One intuitive way to tackle this situation is to simply send all coefficients to the clients and let the clients switch to a suitable model upon its resource allocation, but meanwhile, this induces higher communication costs. It might be also interesting to improve the straggler-resilience of our approach. We leave this extension for future work.

---

> ### Author Response · Authors · 2022-08-02
> **Part 2:  Response to Reviewer f2kP**
>
> **wrt complexity analysis.** Computation complexity is discussed in detail and compared in Sec. 2 (Sec. 3 for the updated version) and Table 2 of the supplementary material. For communication complexity, our method expresses a kernel of shape $k^{2} \times Sp \times Tp$ (with rescale ratio $p$) as a neural basis $V_{\text{share}}$ of shape ${k^{2} \times R_{1} \times R_{2}}$ and coefficients $U_{p}$ of shape ${R_{2} \times Sp/R_{1} \times Tp}$. Thus, the total parameters required by our approach is ${k^{2}R_{1}R_{2}+(R_{2}/R_{1})STp^{2}}$. By properly setting the values of $R_{1}$ and $R_{2}$, we can achieve an on-par or better efficiency compared to existing solutions. We have added further detailed analysis in the supplementary file. An empirical comparison is provided in the last two columns of Table 1 in the original manuscript. We changed the column name to "communication cost" to make it more clear.
>
> For the convergence analysis, we have included an empirical study in Figure 1 of the supplementary file. One can see that our approach has a faster/on-par convergence speed and achieve better accuracy compared to previous solutions.  We agree that theoretical analysis will be definitely helpful. However, we would like to mention that the theoretical convergence analysis is rather onerous in our setting because our method involves optimizing heterogeneous architectures (width), while most previous FL algorithms and theorems only consider shape-aligned models. For the same reason, HeteroFL [16] and FjORD [17] are also not able to conduct such analysis.
>
> **wrt more evidence on knowledge sharing.** As we compose individual weight fragments as linear combinations of the neural basis, it indeed has a similar effect as pruning if all values in the corresponding coefficients (a column of $U_{p}$) are close to zero. To show that our method enables better knowledge sharing, we report the statistics of the mean absolute value for each column in the coefficient matrix. As illustrated in Figure 2 of the supplementary file, values are not small and zero-centered, indicating knowledge stored in the basis is well-shared across models.
>
> **wrt unique complexity levels.** To answer this question, we consider a simulated experiment on CIFAR10 where (1). we double the number of complexity levels (i.e. $p\in \\{ 0.125, 0.25, 0.375, 0.5, 0.625, 0.75, 0.875, 1 \\}$), and (2). each participant in a communication round has a unique complexity level. We report the results in Figure 3 of the supplement file. One can see that our approach can still stably outperforms existing solutions. While it is difficult to conduct an experiment considering a large scale of devices with distinct complexity levels (because, for most ratio $p$,  rescale model width will not result in integer channel numbers), we speculate it would be challenging since the capacity-specific coefficient (only shared among one device) will be biased towards its own data. To this end, the introduced pre-clustering strategy, which quantizes complexity levels, is important to stabilize the training.

---

> ### Author Response · Authors · 2022-08-02
> **Response to Reviewer f2kP**
>
> We thank the reviewer for their constructive feedback and insightful ideas for further improvement.  We hope that our response will address your concerns.
>
>
> **wrt suggested baselines.** We would like to kindly disagree with the statements provided by the reviewer.  The listed references are for very different settings.  Our goal is to tackle federated training for participants with heterogeneous capacity and produce generic global inference models, a very practical scenario described in HeteroFL and FjORD. In this setting, the capacity of each device is collectively constrained by **memory**, **computation budget** and **communication overhead**, which accords with real-world practice, e.g., deployment over multi-generation devices. We model system heterogeneity through rescaling network width by $p$, which directly reduces the communication and computational overhead by $p^2$ and memory consumption by $p$.
>
> **LotteryFL** and **FedMask** focus on a very different problem setting: **1.** as discussed in Sec. 1 of both papers, they aim to learn **personalized** models, a fundamentally different task not aiming for producing global models. **2.** They cannot tackle system heterogeneity. As described in Sec 3.1 in LotteryFL and Sec 3.4 in FedMask, they require all clients to **run the full model** as the initial training stage, indicating a uniform device capacity is required. Hence, their performance is incomparable with us. We have added these papers to the related work.
>
> **FedHM** cannot handle the described heterogeneity setting either, as it lacks the ability to scale memory consumption. As described in Sec 2.1 & 2.2 of FedHM, it factorizes a kernel of shape $m\times n$ (where $m$ and $n$ are input and output channels respectively) into two tensors of shape $m\times r$ and $r\times n$. As such, the network still has the same width $n$ after factorization, indicating it is not able to vary the peak memory consumption (the large intermediate feature maps and their associated gradient computation are the bottlenecks). This makes direct comparison unfair due to unaligned device capacity requirements. In addition, FedHM is a pre-print work without code publicly available to reproduce its results.
>
> **wrt novelty and advantages over previous low-rank methods.** Unlike previous methods, our approach **never performs explicit low-rank factorization**  but instead only uses this notion **conceptually**. Our method significantly differs from previous works in terms of **formulation**, **training strategy** and **problem scope**. In our case, the low-rank nature of deep models serves only as a theoretical foundation for using a unified neural basis to express knowledge and compose heterogeneous models. As described in Sec. 2.2 & 2.4, FLANC directly learns the basis and coefficients from scratch rather than factorizes them from existing kernels.
>
> The proposed neural composition has several advantages compared to previous methods performing real low-rank decomposition: 1. it avoids approximation errors of factorization, which will harm the performance. Meanwhile, it can be applied to every layer for better complexity scaling. As discussed in Sec. 2.2 of FedHM, factorization inevitably introduces **an approximation error** that will be accumulated and propagated to subsequent layers. This makes their approach only applicable to the last several layers and much less practical for real-world deployment, as early layers can also become computational and memory bottlenecks, for example, when dealing with large images. 2. our method can handle system heterogeneity by constructing client models with different widths, whereas factorization methods cannot achieve the same. As discussed, factorization requires the resulting tensors to have aligned input and output dimensions (e.g., $m\times r$ and $r\times n$) with the original kernel (e.g., $m \times n$), which cannot change run-time width to adjust memory consumption. FedDLR and FedPara further assume a uniform processing capacity over devices. These make previous approaches infeasible for dealing with system heterogeneity. We have included this discussion in the supplementary file.

---

> ### Author Response · Authors · 2022-08-06
> **We would like to hear your thoughts about our response to your concerns.**
>
> Given the upcoming OpenReview deadline, we were kindly wondering whether our response and new experiments have addressed your concerns, and we would greatly appreciate a reply.

---

> ### Author Response · Authors · 2022-08-08
> **We look forward to hearing from you.**
>
> Thank you again for your valuable comments. As the discussion period is closing soon, could you please take a look at our response and reevaluate the submission?  Please let us know if there is any further question about the submission. We look forward to hearing from you.

---

### Meta-Review · Area_Chair_QFKE · 2022-08-30

**Recommendation:** Accept
**Confidence:** Less certain

**Metareview:**

This paper proposes a method to cope with heterogeneous computation capabilities of clients in federated learning. The initial reviews were positive, but some the high-score reviewers indicated low confidence. The following concerns were raised.
1. Limitations in the experimental baselines
2. Lack of theoretical justification for the convergence and the communication/computation complexity
3. Somewhat incremental novelty
The authors put in significant effort to address the concerns during the rebuttal which led to a slight increase in the average score. Therefore, I recommend acceptance of the paper. I strongly encourage the authors to take the reviewers' constructive feedback into account when revising the paper.

**Award:**

No

---

### Decision · Program_Chairs · 2022-09-14

Accept